# Quantitative Analysis of the Head Tilt Using Three-Dimensional Temporal Scan in Children with Torticollis

**DOI:** 10.3390/children10020225

**Published:** 2023-01-27

**Authors:** Rayu Yun, Hayoung Jung, Xin Cui, Sungchul Huh, Wonsup Lee, Heecheon You, Sooyeon Kim

**Affiliations:** 1Department of Rehabilitation Medicine, Pusan National University Yangsan Hospital, Yangsan 50612, Republic of Korea; 2Department of Industrial & Management of Engineering, Pohang University of Science and Technology, Pohang 37673, Republic of Korea; 3School of Global Entrepreneurship and Information Communication Technology, Handong Global University, Pohang 37554, Republic of Korea

**Keywords:** torticollis, children, three-dimensional scan

## Abstract

The head tilt of patients with torticollis is usually evaluated subjectively in clinical practice and measuring it in young children is very limited due to poor cooperation. No study has yet evaluated the head tilt using a three-dimensional (3D) scan and compared it with other measurement methods. Therefore, this study aimed to objectively demonstrate head tilt through clinical measurements and a 3D scan in children with torticollis. A total of 52 children (30 males, 22 females; age 4.6 ± 3.2 years) diagnosed with torticollis and 52 adults (26 men, 26 women; age 34.42 ± 10.4 years) without torticollis participated in this study. The clinical measurements were performed using a goniometer and still photography methods. Additionally, the head tilt was analyzed using a 3D scanner (3dMD scan, 3dMD Inc., Atlanta, GA, USA). There was a high correlation between the other methods and 3D angles, and the cut-off value of the 3D angles for the diagnosis of torticollis was also presented. The area under the curve of the 3D angle was 0.872, which was confirmed by a moderately accurate test and showed a strong correlation compared with other conventional tests. Therefore, we suggest that measuring the degree of torticollis three-dimensionally is significant.

## 1. Introduction

Torticollis is a common pediatric musculoskeletal condition [1]. Usually, 3 to 4 months after birth, the child can overcome gravity, move the neck back and forth, and keep the neck vertical. Failure to maintain this posture is called a head and neck posture deviation. Torticollis also collectively refers to postural head and neck abnormalities. It can be classified into torticollis, laterocollis, anterocollis, and retrocollis, depending on its position in the body plane. A left or right rotation on the transverse plane (XZ plane) is called torticollis, a left or right tilt on the coronal plane (XY plane) is called laterocollis, a forward tilted face on the sagittal plane (YZ plane) is called anterocollis, and a backward tilt is called retrocollis [2]. Two or more symptoms may also appear in combination. Therefore, measuring the head deviation from the midline in a two-dimensional manner for the complex symptoms listed above is limited because the head rotation or tilt back and forth cannot be measured.

For adults [3,4,5,6,7,8,9] and older children [10,11,12,13], there are many tools to measure the head deviation, but there are limitations in using these methods for young children. Some studies have compared the passive neck range of motion (ROM) measurement methods in young children [14,15]. In some studies, the active ROM [16] and active or passive side flexion [17] have been evaluated together. In addition, some methods use radiographs [18], photographs [19], or goniometers [20], but there is no gold standard for evaluating head tilt, an objective symptom of patients with torticollis. This is because young children are less cooperative during their examinations [1].

The head deviation from the midline in patients with torticollis is usually evaluated subjectively in clinical practice, and few studies use three-dimensional (3D) evaluation. In one study, the neck ROM was identified in men aged >16 years through 3D analysis [21]; however, we thought that additional evaluation could be made. Furthermore, in patients with torticollis, not only the tilt on the XY plane but also the rotation on the XZ plane and tilting forward or backward on the YZ plane are observed. Therefore, it was considered that 3D evaluation was necessary to perform this evaluation.

This study aimed to objectively demonstrate head tilt through clinical measurements and 3D scanning in young children. With the recent development of 3D scanning technology, it is possible to obtain high-resolution 3D images in a short time; however, no studies have compared the existing evaluation methods by measuring the head deviation from the midline and cervical ROM in 3D in patients with torticollis. Therefore, we performed a quantitative analysis of head tilting using 3D scanning to compare the manual measurements for children with torticollis in Korea. The study was conducted assuming that the overall head deviation angle measured using the 3D scanner differs between normal people from those with torticollis, and has a high correlation with the existing measurement methods.

## 2. Materials and Methods

### 2.1. Design and Participants

This cross-sectional study was conducted between May 2020 and May 2021. The number of samples was determined under the assumption that the overall head deviation angle measured using the 3D scanner would be greater in the patient group than that in the control group, and that there was a positive correlation with other measurement methods. The number of samples was calculated using the G Power 3.1 program. In the comparison between groups, the effect size 0.5, significance level 0.05, and power 80% were set, whereas the effect size 0.6, significance level 0.05, and power 80% were set in the correlation analysis. More than 51 people were targeted in each group [22,23,24]. Among the children who visited the Department of Rehabilitation Medicine at Pusan National University Yangsan Hospital, those diagnosed with torticollis participated in the study. Patients who could not maintain their posture in a sitting position for 1 min or cases where their guardians or patients did not agree with the measurement were excluded. The measurement was explained to all patients and their guardians and proceeded with written consent. A total of 52 children aged 0–18 years (30 males, 22 females; age 4.6 ± 3.2 years) diagnosed with torticollis participated in this study. To set the reference value, 52 adults (26 men, 26 women; age 34.42 ± 10.4 years) without torticollis were set as controls. This study was approved by the Institutional Review Board of Yangsan Pusan National University Hospital, Korea (IRB No. 05–2020-094).

### 2.2. Clinical Measurement

#### 2.2.1. Goniometer Method

Clinical measurements were performed by two physiatrists using a goniometer. The goniometer method measures the slope of the imaginary extension line of the philtrum based on a line parallel to both clavicles, with the goniometer centered on the sternum after placing the patient in a supine position [25] (Figure 1).

#### 2.2.2. Still Photography Method

The patient was placed in a supine position, and pictures were taken using a digital camera by inducing them to look behind the person taking the picture. In the patient’s still photograph, the angle formed by the line connecting the two eyes and the line connecting the acromion was measured [19] (Figure 2). Three photographs were taken to reduce the measurement bias, and the average of the three values was obtained as the final value.

#### 2.2.3. Three-Dimensional Scan Method

Head 3D scanning and data processing were conducted to analyze the head posture of the participants. Before the 3D scanning experiment, circular sticker-type landmarks were attached to the subnasal, sellion, promentale, left/right infraorbitale, left/right ectocanthus, left/right tragion, and left/right acromion of the participant (Figure 3). A 3D head shape, including color texture information, was obtained using a head 3D scanning system (3dMD Inc., GA, USA). The participants in the experiment were controlled to straighten their upper bodies correctly and maintain a forward-looking posture during the 3D scan. The 3D scanning system that creates 3D images of the head consists of a total of five Modular Camera Units (MCUs) and each MCU has 3 machine vision cameras. LED lights are placed in the upper and lower areas of the front, the upper areas of the left and right sides, and the rear areas of the left and right sides. There are 1 MCU in the lower area of the front, 2 MCUs in the left and right upper front corners, and 2 MCUs in the left and right rear corners (Figure 4). The 3D images of the participants were aligned using two acromion joint positions and a lateral axis vector such that the upper body faced forward. Using CAD S/W (Geomagic Design X, 3D Systems Inc., Rock Hill, SC, USA), the 3D coordinates of nine landmarks were stored by referring to the stickers’ position in the 3D scan image, and the midpoint of the left and right tragions and left and right ectocanthi were calculated.

The measurements were made for 20 s using a 3D scanning system, and the machine created a progressive sequence of 3D head images at 10 frames per second. In this study, a direction vector for head deviation measurement was calculated using each 3D head image taken, and the most frequent value among the measured values was used for analysis.

The reference vectors in a neutral posture for comparison with the direction vectors representing the head deviation of the participants were generated using 3D head shapes and landmark information. Vertical vectors (0, 1, and 0) were created as a reference for measuring the head lateral flexion/extension angle (roll angle). An anterior vector (0, 0, and 1) was created as a reference for measuring the head flexion/extension angle (pitch angle) and rotation angle (yaw angle). The neutral position of the head in terms of the head flexion/extension was defined as an atomic ear-eye line 15° upward relative to the horizontal [26]. For the reference vector representing the overall 3D deviation of the head (3D angle), a diagonal vector (0, 0.707, and 0.707) was generated (Figure 5).

A direction vector for measuring the roll angle of the head was defined as a unit vector passing through the sellion and promentale. A direction vector for measuring the yaw angle of the head was defined as a unit vector passing through the midpoint of the tragions and subnasale. A direction vector for measuring the pitch angle of the head was defined as a unit vector passing through the midpoint of the tragions and the midpoint of the ectocanthi. A direction vector for measuring the 3D angle was defined as a unit vector of the sum of the vertical and anterior vectors (Figure 6).

The head angles were analyzed by measuring the pitch angle, roll angle, and yaw angle using the dot products of the reference and direction vectors for the head flexion/extension, head lateral flexion/extension, and rotation side, respectively (Figure 7). In the case of the yaw angle, the case with the right rotation was expressed as a positive number and the left rotation as a negative number. In the case of the roll angle, the right tilt was expressed as a positive and the left tilt as a negative. In the case of the head pitch angle, the case of the upward compared to the horizontal was expressed as a positive and the downward as a negative. The 3D angle representing a comprehensive head deviation was calculated using the dot product of the diagonal vectors of the reference and head scans, and was expressed only as a positive.

### 2.3. Statistical Analysis

The comparison of goniometer measurements between the two inspectors, mean of goniometer measurement and still photography measurements, and the mean of the goniometer measurement and yaw angle measured using a 3D scanner were calculated using the intra-class correlation coefficient (ICC). Pearson’s correlation analysis was conducted to analyze the correlation between the mean of the goniometer measurements and the 3D angle using a 3D scanner. The Mann–Whitney U test was performed for the average comparison of the 3D angle using a 3D scanner between patients with torticollis and the control groups. The significance level was set at *p* < 0.05. A receiver operating characteristic (ROC) curve analysis was performed to determine the cut-off value of the 3D angle for the torticollis diagnosis. In the ROC curve, when the sensitivity and specificity were displayed on a linear chart, the point at which the two graphs met was set as the cut-off value.

## 3. Results

### 3.1. Angles with Multiple Methods

The values in the control and torticollis groups were obtained using a goniometer, still photography, and a 3D scanner. Since measured values, except the 3D angle, have positive and negative values depending on the direction, there may be an error in calculating the average and standard deviation, so all values were converted into absolute values and their average and standard deviation were analyzed. The mean and standard deviation of the absolute values between the groups of measured values obtained by each method are presented in Table 1.

The average angles between the two groups were compared. The existing measurement, goniometer, and still photography methods showed significant differences between the two groups. In addition, the yaw, roll, pitch, and 3D angles showed statistically significant differences between the two groups (*p* < 0.001).

### 3.2. Reliability of Torticollis Measurement in Goniometer Method

The ICC (3,1) was performed to measure the absolute agreement of the values measured by the goniometer method by two physiatrists, and the ICC value was 0.997 [95% confidence interval (CI) = 0.995–0.998, *p* < 0.001] (Table 2).

### 3.3. Reliability of Torticollis Measurement in Goniometer and Still Photography Methods

The ICC (3,1) was performed to measure the absolute agreement between the goniometer method by physiatrist one and the still photography method, and between goniometer method by physiatrist two and still photography method. The ICC values were 0.996 [95% CI = 0.994–0.997, *p* < 0.01] and 0.997 [95% CI = 0.996–0.998, *p* < 0.001] (Table 2).

### 3.4. Reliability of Torticollis Measurement in Goniometer Method and Roll Angle Using the 3D Scanner

The ICC (3,1) was used to measure the absolute agreement between the goniometer method by physiatrist one and the roll angle with the 3D scanner, and between the goniometer method by physiatrist two and the roll angle with the 3D scanner. The ICC values were 0.944 [95% CI = 0.918–0.962, *p* < 0.001] and 0.949 [95% CI = 0.925–0.965, *p* < 0.001] (Table 2).

### 3.5. Correlation between the Goniometer Method and 3D Angle

The 3D angles were expressed only by positive values. Accordingly, the relationship between the average of the absolute value of the measured values using the goniometer method and the 3D angle was confirmed, and strong positive correlations were confirmed with an analysis coefficient of 0.720 (*p* < 0.001) (Figure 8).

### 3.6. Correlation between Still Photography Method and 3D Angle

The relationship between the average of the absolute value of the measurements of the still photography method and the 3D angle was confirmed, and strong positive correlations were confirmed, with a correlation coefficient of 0.727 (*p* < 0.001) (Figure 9).

### 3.7. Cut-Off Value of 3D Angle for Diagnosis of Torticollis

For the diagnosis of torticollis using the measured 3D angle, the cut-off value was obtained through the ROC curve, the AUC was 0.872, the cut-off value was 4.950, the sensitivity was 0.788, and the 1-specificity was 0.212 (Figure 10).

## 4. Discussion

Congenital muscular torticollis is the most common cause of torticollis in infants and children. While the prevalence varies depending on the report, the prevalence is reported to be 0.3% to 2%, and approximately 70% of the head and neck position abnormalities seen during childhood are congenital muscle torticollis [27]. In muscular torticollis, one side of the sternocleidomastoid muscle is shortened, the head is tilted laterally to the affected side and rotated to the other side, and rotational restriction to the affected side appears [28,29,30]. A limitation of rotational ROM that does not improve despite long-term appropriate physical therapy is considered an indication for the surgical treatment of congenital muscular torticollis [31]. Therefore, when a child with suspected congenital muscle torticollis visits the hospital, the physician should assess the head deviation from the midline and the rotational component. However, the quantitative assessment of these factors in infants and young children is difficult, and standardized assessments have not yet been established. If congenital muscular torticollis is not treated in a timely manner and the head deviation from the midline and rotational limitation persist, secondary plagiocephaly, facial asymmetry, scoliosis, and in very severe cases, spinal cord damage can occur, requiring the prompt diagnosis and treatment [18,32].

Head deviation from the midline in patients with torticollis is usually evaluated subjectively in clinical practice. Measuring the head deviation from the midline can be performed using a goniometer in the clinic [20] or by taking still photographs [19,33] or radiographs [18]. A study using still photography to measure the midline head deviation of infants showed high intra-rater reliability (ICC 0.79–0.84) and fair to high inter-rater reliability (ICC 0.72–0.99) [19]. To measure the head deviation from the midline on a plain radiograph of the cervical spine, the angle between the line connecting the mastoid processes on both sides and the line connecting the center of the 3–7 cervical spinous processes was measured [34]. There are several methods to measure the head deviation from the midline, but no standardized method exists. In addition, it is very difficult for children to constantly maintain the same posture for measurement or keep their eyes backward during measurement. Owing to these limitations, measuring the head deviation from the midline and objective changes is difficult.

This study is the first to obtain not only the head tilting angle but also the left, right rotation, vertical, and combined 3D angles using a 3D scanner. Previous studies have used 3D scanners to confirm facial asymmetry in patients with torticollis [35], but this is the first in which a new angle has been presented.

The reliability between the measures was confirmed using the goniometer method and between the goniometer and still photography methods. In addition, high reliability was confirmed between the yaw angle obtained using the 3D scanner and other methods. There was a high correlation between other methods and 3D angles, and the cut-off value of 3D angles for the diagnosis of torticollis was also presented. The AUC of the 3D angle was 0.872, confirmed by a moderately accurate test, and showed a strong correlation compared with that of other conventional tests [36,37]. Given that highly trained multi-year-old professionals have conducted the existing tests, we believe the 3D angle is sufficient for clinical use. Thus, we suggest that measuring the degree of torticollis three-dimensionally is significant.

This study had some limitations. Most studies of patients with torticollis target only one group according to causes, such as muscular, developmental, or ocular torticollis. It is well known that the rotational limitations are important factors in the recovery of patients with muscular torticollis and determine the need for surgical treatment in the future. However, there are no studies on whether the rotational components and up and down restriction affect the course of treatment in children with torticollis due to other causes, such as muscular torticollis. Accordingly, further research will confirm whether these components affect treatment progress in patients with torticollis due to causes other than muscular torticollis.

Another limitation is that, as in other studies with children, children with difficulty maintaining posture were excluded from the study. However, the difference in this study is that when analyzing an angle using a 3D scanner, the angle analysis was performed by photographing several frames per second, and the angle was determined using the most frequent values. In measurement studies using photographs and measurement studies using an angle meter, there is a limitation that there may be some differences between the measurements because only the angle during the measurement, which is a very short time, can be measured.

Third, people with torticollis were measured for children under the age of 18, but the measurement for people without torticollis was conducted for general adults, and there was a difference in the age between the two groups. However, the examination in the adult patients showed the distribution of yaw, roll, and pitch angle values in people without torticollis, and for this, they had to be measured in people with high cooperation. Considering that the average age of the patient group with torticollis was 4.6 years old, it is difficult to obtain an accurate reference value because of the difficulty in cooperation when measuring using children without torticollis of a similar age. It is thought that more data by age can be collected through further research in the future.

Fourth, the 3D scanner used in the study was very expensive, making it difficult to use in a medical center that treats patients with torticollis. However, even without using a 3D scanner, it is thought that recognizing certain parts of the face and body can obtain a similar angle through several photos. Through continuous research in the future, it will be possible to obtain 3D angles using several photos. If this occurs, the degree of torticollis can be measured in 3D in clinical practice without the need for expensive 3D equipment.

Fifth, we measured the left and right head tilts of torticollis patients with the goniometer method and the still photography method and confirmed the correlation between the head tilt angle and the roll angle and 3D angle measured with a 3D scanner. However, the head rotation angle or head flexion/extension angle was measured only with a 3D scanner, not by the other direct measurement methods. If these values were measured together and the correlation with the yaw angle and pitch angle measured with the 3D scanner were also confirmed, it would have helped to increase the reliability of the measurement method using the 3D scanner. Therefore, additional research is needed in the future. Lastly, the temporal 3D scanner used in our study is a device that can obtain 3D images dynamically, having many advantages compared with evaluations in a static condition. It can be used even when a particular posture is difficult to maintain due to frequent posture changes among children and the range of motion needs to be analyzed. Using the temporal 3D scanner, we are planning future research to examine the improvement of children with torticollis and the effect of the limitation of the range of motion on improvement.

## 5. Conclusions

The present study evaluated torticollis using a three-dimensional head scan, which has not been tried before, proposed the new concept of the 3D angle measurement for torticollis, and evaluated its effectiveness in the clinical context. Compared with that of the control group, the 3D angle value was significantly larger in the patient group, and there was a high correlation with other existing torticollis measurement methods. Thus, it is thought that torticollis can be evaluated three-dimensionally, and further research is needed to use this value in clinical practice.

## Figures and Tables

**Figure 1 children-10-00225-f001:**
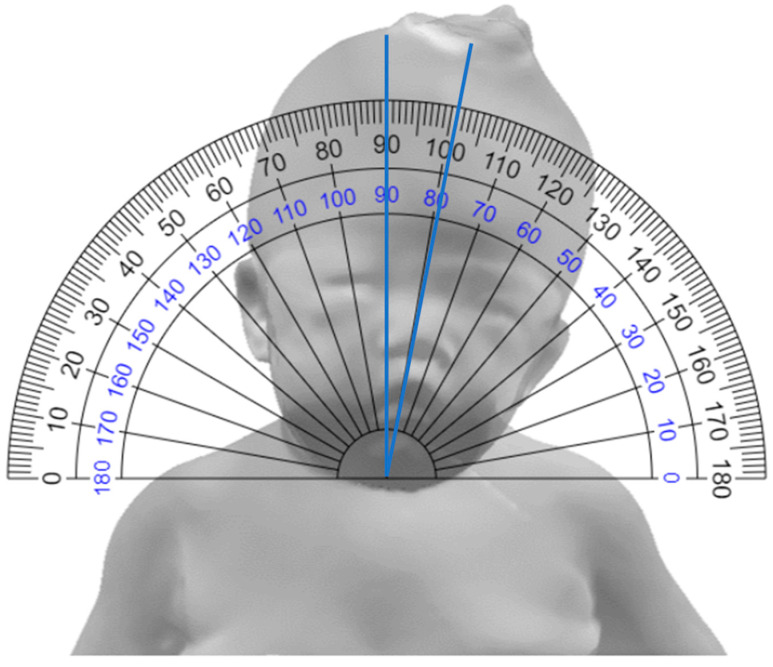
Goniometer method.

**Figure 2 children-10-00225-f002:**
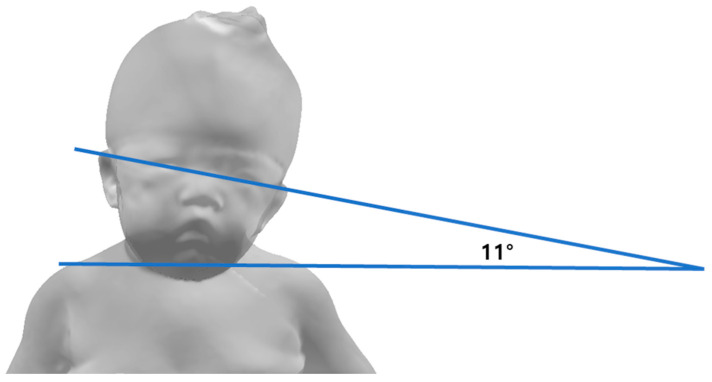
Still photography method.

**Figure 3 children-10-00225-f003:**
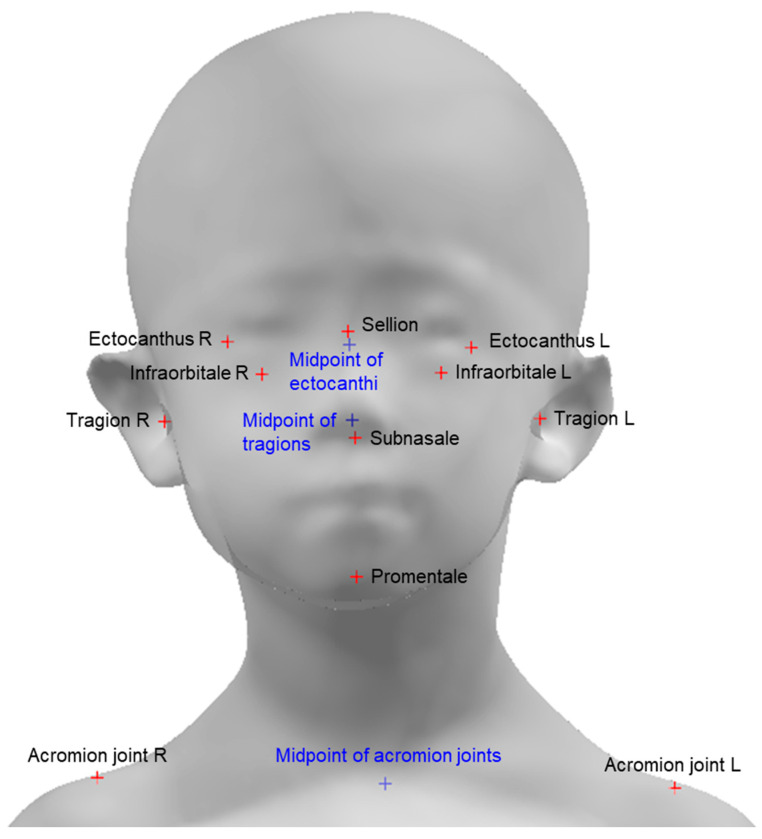
Landmarks for head angle analysis.

**Figure 4 children-10-00225-f004:**
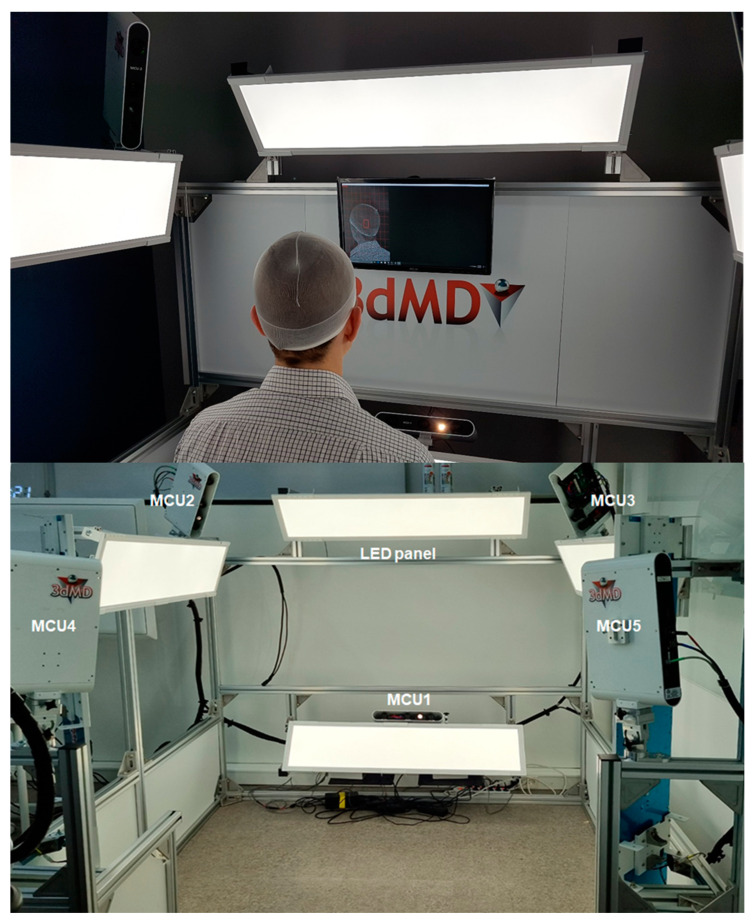
Three-dimensional head scanning using 3dMD.

**Figure 5 children-10-00225-f005:**
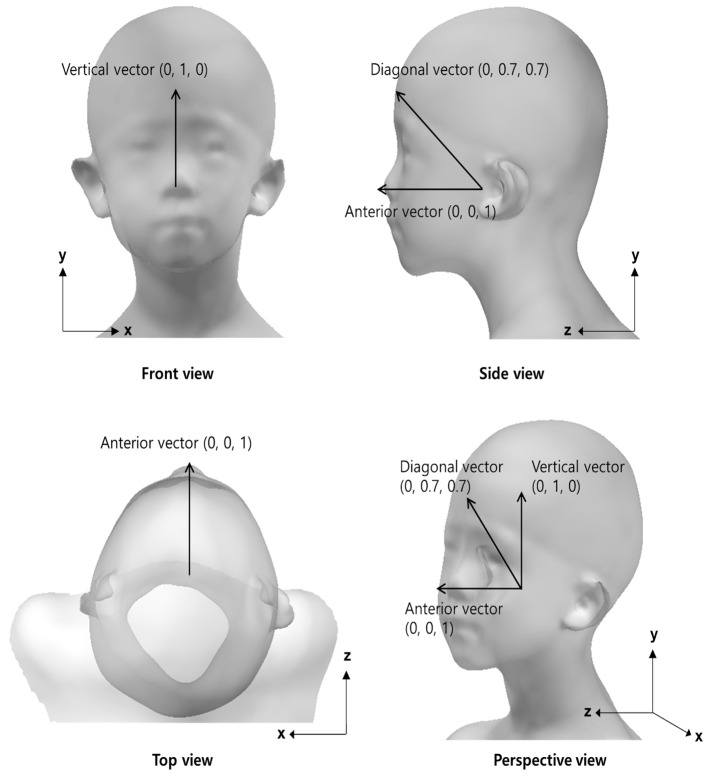
Reference vectors for measurement of head angles.

**Figure 6 children-10-00225-f006:**
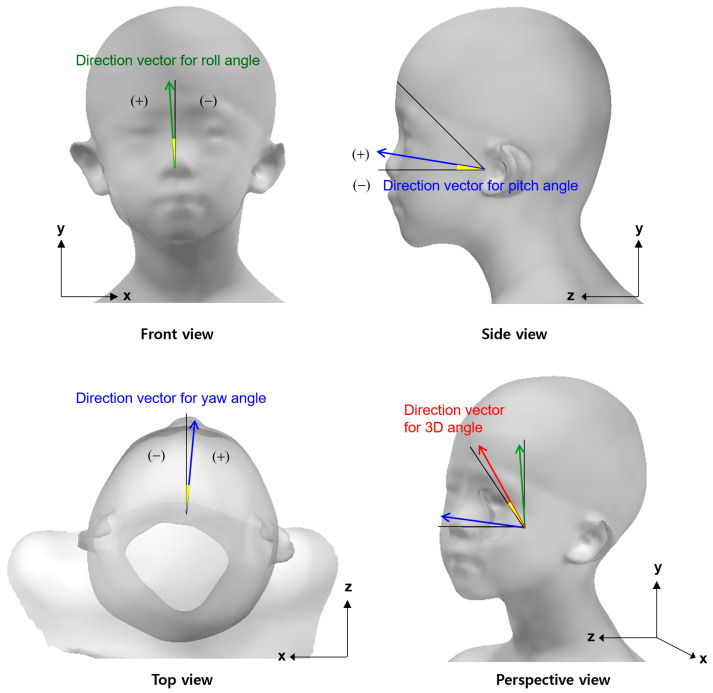
Direction vectors for measurement of head deviation compared to reference vectors.

**Figure 7 children-10-00225-f007:**
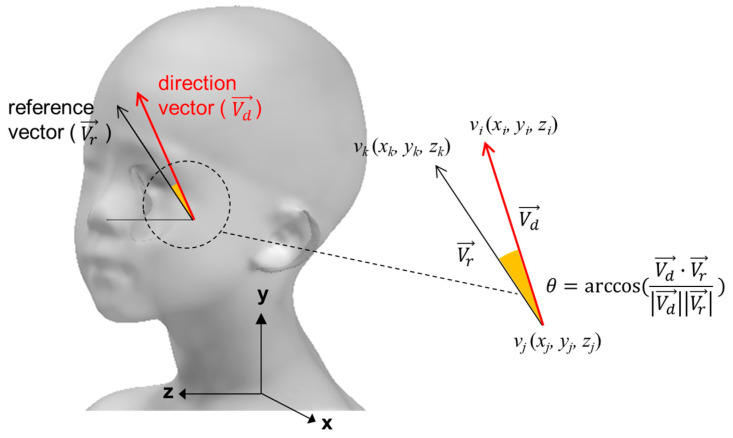
The 3D angle between reference vector and direction vector.

**Figure 8 children-10-00225-f008:**
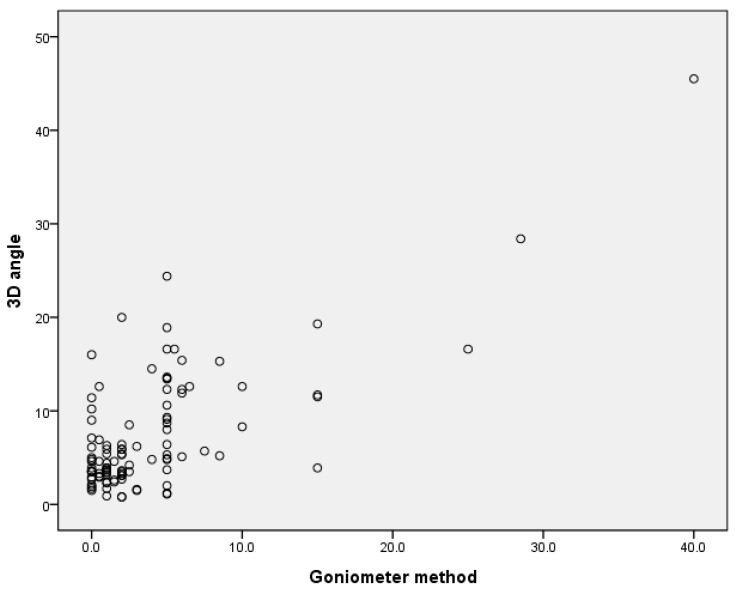
Correlation between the goniometer method and 3D angle.

**Figure 9 children-10-00225-f009:**
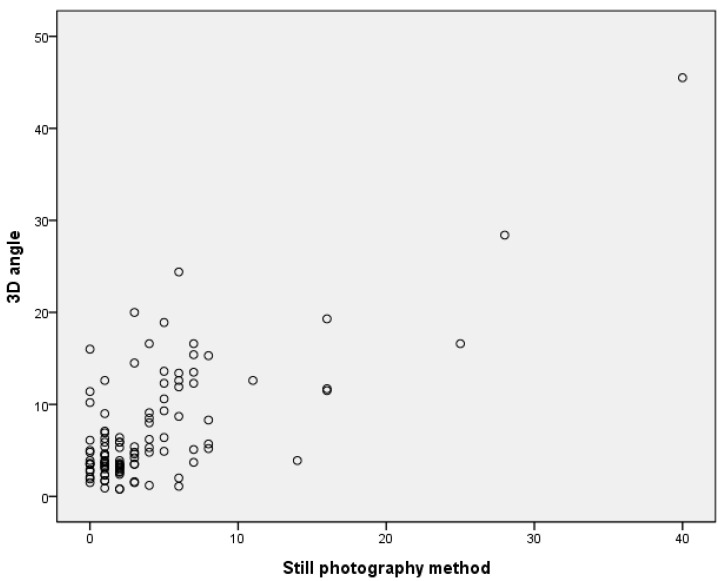
Correlation between still photography method and 3D angle.

**Figure 10 children-10-00225-f010:**
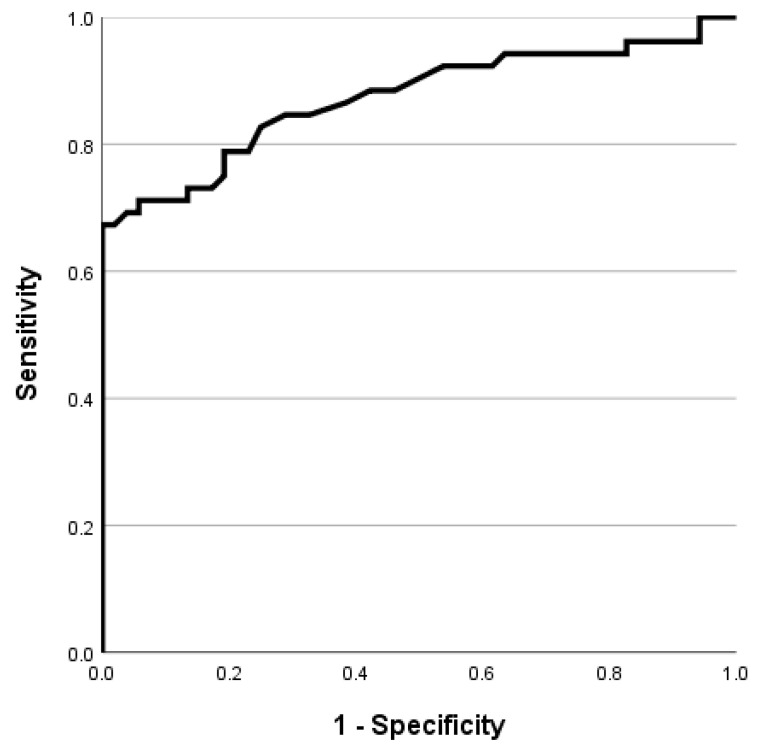
ROC curve of 3D angle.

**Table 1 children-10-00225-t001:** Angles with multiple methods and difference in angles between the control and torticollis groups.

	Control Group	Torticollis Group	*p*-Value
Goniometer method by physiatrist 1	1.13 ± 1.10	6.67 ± 7.45	<0.01
Goniometer method by physiatrist 2	1.15 ± 1.02	6.77 ± 7.24	<0.01
Still photography method	1.38 ± 1.07	6.98 ± 7.26	<0.01
Yaw angle with 3D scanner	1.54 ± 0.91	6.08 ± 5.97	<0.01
Roll angle with 3D scanner	2.13 ± 1.60	6.19 ± 6.33	<0.01
Pitch angle with 3D scanner	5.78 ± 3.52	10.37 ± 6.65	<0.01
3D angle (comprehensive angle considering yaw, roll, and pitch angles)	3.60 ± 1.55	11.07 ± 7.65	<0.01

Values are presented as means of absolute values ± standard deviation.

**Table 2 children-10-00225-t002:** Reliability of torticollis measurements.

	ICC (3,1)	95% CI	*p*-Value
Goniometer method of two physiatrists	0.997	0.995–0.998	<0.001
Goniometer method by physiatrist 1 and still photography method	0.996	0.994–0.997	<0.001
Goniometer method by physiatrist 2 and still photography method	0.997	0.996–0.998	<0.001
Goniometer method by physiatrist 1 and roll angle with 3D scanner	0.944	0.918–0.962	<0.001
Goniometer method by physiatrist 2 and roll angle with 3D scanner	0.949	0.925–0.965	<0.001

## Data Availability

The data presented in this study are available on request from the corresponding author. The data are not publicly available due to privacy or ethical restrictions.

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
