# Peer review of "Quantitative Analysis of the Head Tilt Using Three-Dimensional Temporal Scan in Children with Torticollis"

_children, 2023, doi:10.3390/children10020225_

Round 1

Reviewer 1 Report

This paper is to investigate the usefulness of clinical evaluation using 3D scanner for torticollis patients.

Since torticollis is a concept that includes not only head tilting but also head rotation. Does the 3D angle calculated by measuring with a 3D scanner represent the sum of these 2-axis deformations? If so, it is desirable to express the current 3D angle as the sum of the deformations of the two axes.

Also, you only looked at the correlation between the head tilting value measured by the goniometer and the roll angle by 3D. Of course, this method seems very reasonable. However, there is no clinically measured data for the rotation angle, and only the head tilting value was investigated for its correlation with the 3D angle. It seems that this point should be inserted in the limitations of the study.

Author Response

Point 1: This paper is to investigate the usefulness of clinical evaluation using 3D scanner for torticollis patients.

Since torticollis is a concept that includes not only head tilting but also head rotation. Does the 3D angle calculated by measuring with a 3D scanner represent the sum of these 2-axis deformations? If so, it is desirable to express the current 3D angle as the sum of the deformations of the two axes.

Response: Thanks for your kind reminders. As you mentioned, torticollis includes not only head tilting but also head rotation. In addition to this, head flexion and extension posture were also considered. Therefore, we obtained the roll angle with a vertical vector for left and right tilting, the pitch angle with a diagonal vector for flexion extension, and the yaw angle with an anterior vector for head rotation. This is also illustrated in clinical measurement 2.2.3 and figure 5. So, the value we express as 3D angle is a value that considers all 3-axis (head flexion/extension, head lateral flexion/extension, and rotation). There may be confusion because of the “3D angles between two vectors”, which is a picture description attached to Figure 6, so we changed it to "3D angle between reference vector and direction vector", and to make it more clear, we changed it to a new picture using the perspective view picture shown in figure 5. The meaning of figure 6 and 3D angle is the value between the direction vector considering 3-axis and the reference vector.

We modified Figure 6, which is included in the attachment.

Point 2: Also, you only looked at the correlation between the head tilting value measured by the goniometer and the roll angle by 3D. Of course, this method seems very reasonable. However, there is no clinically measured data for the rotation angle, and only the head tilting value was investigated for its correlation with the 3D angle. It seems that this point should be inserted in the limitations of the study.

Response: Thanks for your kind reminders. As you mentioned, if the rotation angle or head flexion/extension angle were measured directly and compared with the values measured using a 3D scanner, the reliability of the 3D angle would have been further improved. The part you mentioned has been added to the limitation as follows, and further research is planned using it.

“Fifth, we measured the left and right head tilts of torticollis patients with the goniometer method and the still photography method, and confirmed the correlation between this and the roll angle and 3D angle measured with a 3D scanner, but the head rotation angle or head flexion/extension angle was measured only with a 3D scanner and was not measured by other direct measurement methods. If these values were measured together and the correlation with the yaw angle and pitch angle measured with the 3D scanner was also confirmed, it would have helped to increase the reliability of the measurement method using the 3D scanner. Therefore, additional research will be conducted in the future. “[Pg12, Ln282-290]

Lastly, thank you very much for giving us a good opinion to develop our paper once again.

Thank you in advance for your kind consideration, and I am looking forward to hearing a positive reply from you soon.

Reviewer 2 Report

I appreciate your work in developing a useful assessment for clinical practice and research.

Author Response

Point 1: I appreciate your work in developing a useful assessment for clinical practice and research.

Response 1: First of all, thank you very much for reviewing our paper. We will try to continue the related follow-up research based on the limitation we wrote.

Reviewer 3 Report

I do congratulate for the novel research in quantitative analysis of the head tilt in children torticolis. The Introduction, Methods, Results, Discussion and Conclusion are well defined. Nevertheless the lack of confirmation about possible clinical use of method is mentioned correctly in the Conclusion.

Author Response

Point 1: I do congratulate for the novel research in quantitative analysis of the head tilt in children torticolis. The Introduction, Methods, Results, Discussion and Conclusion are well defined. Nevertheless the lack of confirmation about possible clinical use of method is mentioned correctly in the Conclusion.

Response 1: First of all, thank you for reviewing our paper. There are many limitations, but we will try to continue the relevant follow-up studies based on the limitations we have written. Thanks once again.
